# Shock Index Is a Validated Prediction Tool for the Short-Term Survival of Advanced Cancer Patients Presenting to the Emergency Department

**DOI:** 10.3390/jpm12060954

**Published:** 2022-06-10

**Authors:** Zhong Ning Leonard Goh, Mu-Wei Chen, Hao-Tsai Cheng, Kuang-Hung Hsu, Chen-Ken Seak, Joanna Chen-Yeen Seak, Seng Kit Ling, Shao-Feng Liao, Tzu-Heng Cheng, Yi-Da Sie, Chih-Huang Li, Hsien-Yi Chen, Cheng-Yu Chien, Chen-June Seak, SPOT Investigators

**Affiliations:** 1Sarawak General Hospital, Kuching 93586, Sarawak, Malaysia; lgzn92@gmail.com (Z.N.L.G.); joannaseak@hotmail.com (J.C.-Y.S.); 2Department of Emergency Medicine, New Taipei Municipal Tucheng Hospital, New Taipei City 23652, Taiwan; b9502086@cgmh.org.tw; 3Department of Emergency Medicine, Lin-Kou Medical Center, Chang Gung Memorial Hospital, Taoyuan 33305, Taiwan; da86003@gmail.com (M.-W.C.); khsu@mail.cgu.edu.tw (K.-H.H.); joeliao76@gmail.com (S.-F.L.); y17322@cgmh.org.tw (C.-H.L.); hshychen@gmail.com (H.-Y.C.); rainccy217@gmail.com (C.-Y.C.); 4College of Medicine, Chang Gung University, Taoyuan 33302, Taiwan; howardandbetty@yahoo.com.tw; 5Department of Gastroenterology and Hepatology, New Taipei Municipal Tucheng Hospital, New Taipei City 23652, Taiwan; 6Department of Gastroenterology and Hepatology, Lin-Kou Medical Center, Chang Gung Memorial Hospital, Taoyuan 33305, Taiwan; 7Graduate Institute of Clinical Medicine, College of Medicine, Chang Gung University, Taoyuan 33302, Taiwan; 8Healthy Aging Research Center, Chang Gung University, Taoyuan 33302, Taiwan; 9Laboratory for Epidemiology, Department of Health Care Management, Chang Gung University, Taoyuan 33302, Taiwan; 10Research Center for Food and Cosmetic Safety, College of Human Ecology, Chang Gung University of Science and Technology, Taoyuan 33303, Taiwan; 11Department of Safety, Health and Environmental Engineering, Ming Chi University of Technology, Taoyuan 243303, Taiwan; 12Queen Elizabeth Hospital, Kota Kinabalu 88586, Sabah, Malaysia; jonathanseak@gmail.com (C.-K.S.); jacklynseng86@gmail.com (S.K.L.); 13Department of Emergency Medicine, China Medical University Hospital, Taichung 404332, Taiwan; easythinking20@gmail.com; 14Department of Emergency Medicine, Ton-Yen General Hospital, Zhubei 30268, Taiwan

**Keywords:** shock index, advanced cancer, emergency physicians, emergency department, 60-day survival, Stratification to Prevent Overcrowding Taskforce (SPOT)

## Abstract

Advanced cancer patients who are not expected to survive past the short term can benefit from early initiation of palliative care in the emergency department (ED). This discussion, however, requires accurate prognostication of their short-term survival. We previously found in our retrospective study that shock index (SI) is an ideal risk stratification tool in predicting the 60-day mortality risk of advanced cancer patients presenting to the ED. This study is a follow-up prospective validation study conducted from January 2019 to April 2021. A total of 410 advanced cancer patients who presented to the ED of a medical centre and could be followed-up feasibly were recruited. Univariate and multivariable logistic regression analyses were performed with receiver operator calibrating (ROC) curve analysis. Non-survivors had significantly lower body temperatures, higher pulse rates, higher respiratory rates, lower blood pressures, and higher SI. Each 0.1 increment of SI increased the odds of 60-day mortality by 1.591. Area under ROC curve was 0.7819. At optimal cut-off of 0.94, SI had 66.10% accuracy. These results were similar to our previous study, thus validating the use of SI in predicting the 60-day mortality of advanced cancer patients presenting to the ED. Identified patients may be offered palliative care.

## 1. Introduction

Patients with advanced cancer account for an increasing number of emergency department (ED) visits, due to an expanding elderly population as well as improved post-diagnosis lifespans with the advancement of cancer therapies [1]. For these patients, aggressive life-sustaining interventions initiated in the ED have been found to be associated with minimal gains in post-admission survival, without significant differences in overall survival or quality of life [2,3]. Advanced cancer patients consequently face increased suffering for the remainder of their lives while their families are saddled with the financial burden of huge hospital bills [4,5]; this occurrence is especially true in countries with limited health insurance systems. As such, the idea of initiating palliative care for these patients right from the start in the ED was mooted, and it has been shown to improve quality of life without adversely impacting survival rates [6].

Prior to initiating palliative care, the emergency physician (EP) and other attending clinicians in the ED ideally should have a means to estimate the short-term survival of each individual patient with advanced cancer. Various retrospective studies have, however, shown that subjective prognostication by doctors were largely imprecise and inaccurate [7,8,9]. Several scoring systems were then studied to objectively evaluate short-term survival rates between one to six months, though the complexity of these scores meant that their utility was limited in the ED environment [10,11,12,13,14,15,16].

A study by Llobera et al. found that terminal cancer patients had a median survival of 59 days [17]. If advanced cancer patients presenting to the ED are unlikely to survive past 59 days, it then stands to reason that they should be provided early with the option of palliative care services. Based on this, we embarked on a retrospective study that found shock index (SI) to be an ideal tool in predicting the 60-day mortality risk of advanced cancer patients presenting to the ED [18]. SI is defined as the ratio of pulse rate to systolic blood pressure [19] and has been widely studied in the prognostication of pneumonia [20,21,22], influenza [23], Coronavirus disease 2019 [24], acute pulmonary embolism [25], acute myocardial infarction [26,27], stroke [28], and trauma [29,30].

Following the positive results from our prior retrospective study, we decided to follow-up with this current study to prospectively validate the use of SI in predicting the 60-day mortality of advanced cancer patients presenting to the ED.

This study is part of a series by the Stratification to Prevent Overcrowding Taskforce (SPOT) investigators, a research group dedicated to maximising clinical outcomes right from the ED via rapid and accurate identification of patients requiring urgent intervention, with the secondary objective of alleviating ED overcrowding. We have to date studied several risk stratification tools in intra-abdominal infections [31,32,33,34,35,36], snakebites [37], and now advanced cancer [18], amongst others [38,39].

## 2. Materials and Methods

### 2.1. Study Design

This prospective observational study was conducted in the ED of Linkou Chang Gung Memorial Hospital (3406 beds with approximately 15,000 ED visits monthly in 2019), the largest tertiary centre in Taiwan [40,41]. This study was approved by Chang Gung Medical Foundation Institutional Review Board (IRB No. 201900493B0). Written informed consent was obtained from all patients and/or legal guardians.

### 2.2. Setting and Subjects

All adult advanced cancer patients above the age of 18 years who visited the ED of our hospital from January 2019 to April 2021 were invited to participate in this study, with the explicit understanding that the research data obtained would not be used to influence decisions on management options and goals. All patients received prompt treatment for their respective presenting illnesses as per our ED protocols. Advanced cancer was defined as locally recurrent or metastatic solid cancer that cannot be cured [42,43,44]. All recruited patients were followed till death or end of study. Any patients lost to follow-up were excluded in the final analysis.

### 2.3. Measurement of Variables

The SI is calculated by dividing the pulse rate by systolic blood pressure. These calculations were performed by a general practitioner blinded to the study objectives. Our primary outcome was short-term survival, defined as survival of 60 days after ED presentation. The study endpoint was taken at 60 days post-ED presentation or mortality.

### 2.4. Statistical Analysis

Continuous variables were presented as mean ± SD while categorical variables were expressed as frequencies (%), with statistical analyses performed with independent sample Student’s t-test and chi-squared test, respectively. Multivariable logistic regression was subsequently carried out to obtain the odds ratio with respect to 60-day mortality, and receiver operator calibrating (ROC) curve of this study population was plotted. Validation of our previous study’s cut-off point of 0.94 was performed via evaluation of its sensitivity, specificity, negative predictive value, positive predictive value, and accuracy in this current study population. Kaplan-Meier analysis was also employed to examine survival between groups with high versus low SIs. *p*-values of <0.05 were taken to be statistically significant. 

## 3. Results

A total of 410 advanced cancer patients were recruited during the study period. Comparison of patient characteristics of survivors versus non-survivors revealed that non-survivors had a significantly higher proportion of patients with hepatocellular carcinoma, as well as a significantly lower proportion of patients with history of prior surgical intervention (Table 1).

In terms of clinical presentation, univariate analysis found the following significant findings: non-survivors had lower body temperatures, higher pulse rates, higher respiratory rates, as well as lower systolic, diastolic, and mean arterial blood pressures compared to survivors. Mean SI of non-survivors was also significantly higher than that of survivors (1.19 versus 0.87) (Table 2).

The aforementioned variables with statistically significant differences further underwent a backward model selection process using multiple logistic regression analysis. SI was found to be the only variable that was significantly related to 60-day survival. After adjusting for age and gender, each 0.1 increment of SI increased the odds of mortality within 60 days of ED presentation by a factor of 1.591 (95% CI: 1.42–1.78; *p* = 0.0012). Area under ROC curve was found to be 0.7819 (Figure 1).

Validation of our previous study’s optimal cut-off point of 0.94 in this current study population found that it had a comparably good performance, with sensitivity 73.65%, specificity 61.83%, positive predictive value of 52.15%, negative predictive value of 80.60%, and accuracy 66.10% (Table 3). Patients with SIs > 0.94 had a hazard ratio of 3.442 compared to those with SIs < 0.94 (*p* < 0.0001).

Kaplan-Meier curve analysis revealed that the 60-day mortality in advanced cancer patients with SI > 0.94 was significantly higher than those with lower SI (Figure 2).

## 4. Discussion

Predicting short-term survival of cancer patients is challenging. Several methods of estimating survival rates have been studied, though with varying accuracies. Even when detailed records of cancer patients’ clinical progression and treatment history were made available, physicians could predict 180-day mortality accurately only three out of four times [14]. With comparable accuracy rates of 73.11% in our retrospective study and 66.10% in this current validation study, SI is therefore a powerful risk stratification tool for rapid prognostication of 60-day mortality in advanced cancer patients presenting to the ED [18]. 

The findings in our current study closely mirror those from our previous retrospective study–SI remained the only significant predictor of 60-day mortality after application of multiple logistic regression analysis. Further validation of our previous study’s cut-off point of 0.94 found that it was still able to identify 73.65% of patients who might benefit from early initiation of palliative care. Nevertheless, it must be heavily emphasized that SI should not be taken as the sole deciding factor in determining goals of therapy, but rather as an adjunct to the ongoing conversation with the cancer patient and family about their wishes regarding end-of-life care.

The beauty of SI lies in its simplicity of calculation, based on two vital sign measurements which can be rapidly obtained in less than a minute. With an optimal cut-off point of 0.94, clinicians in the ED should consider discussing with advanced cancer patients and their families regarding the option of palliative care once they see that pulse rate readings are almost equal to or higher than the corresponding systolic blood pressures.

The accuracy of SI in predicting 60-day mortality in this patient population is because of its association to performance status of the circulatory system. Circulatory failure is often implicated in the death of advanced cancer patients, due to a combination of generalized cachexia, cardiac cachexia, and anorexia leading to poor nutrition and dehydration [18]. This deterioration in cardiac function is consequently reflected as elevated SI in advanced cancer patients.

Accurate estimation of survival is vital for effective palliative care [45]. Early palliative care has also been demonstrated to significantly improve quality of life as compared to standard care [46]. Clinicians are, however, frequently inaccurate in their predictions of patient survival, often overestimating their patients’ remaining lifespan [47,48,49,50]. This subsequently limits advanced cancer patients’ access to palliative care [51]. The use of SI in prognosticating these patients in the ED thus has the potential to improve patient care by providing them and their families with a more accurate estimation of their 60-day survival [52]. Junior doctors will be empowered to initiate conversations regarding end-of-life care and advance medical directives with patients and their families right at the start of the patient encounter in the ED [53]. This can then be followed by more in-depth discussions with the patients’ primary attending oncologists.

Such an approach is especially useful in scenarios where these patients present during out-of-office hours when oncology services are not readily available in the same or different medical centre. After initial counselling for palliative care by ED doctors for patients with high SIs, the patients and their families can take their time to discuss matters amongst themselves; once they have agreed to further consultations with the palliative care team, referrals can be made accordingly at the start of the next day shift. If the suggestion is rejected outright, the ED team would then be able to proceed with their usual curative management. The application of SI can therefore potentially enable identified patients to benefit from early palliative care, while having minimal increase in after-hours hospice referrals.

Again, it is important to note that discussions surrounding end-of-life care are complex and involve a lot of stakeholders. SI should not be used as the sole determining factor in justifying an abandonment of all curative treatment in favour of palliative therapy. Rather, SI is a tool in identifying ED patients who are likely to benefit more from palliative care as opposed to aggressive interventions. Subsequent management should depend on discussions between medical teams and patients with their families.

The findings of this current validation study, together with those of our previous study [18], successfully demonstrates SI as an ideal risk stratification tool for predicting the 60-day mortality risk of advanced cancer patients presenting to the ED. Further studies can look into the applicability of SI in other terminal illnesses.

## 5. Conclusions

Shock Index is an ideal risk stratification tool for predicting the 60-day mortality risk of advanced cancer patients presenting to the ED. Clinicians working in the ED should use SI to rapidly identify patients who are likely to benefit more from palliative care as opposed to aggressive intervention. Open discussion regarding end-of-life care can then be initiated with these identified advanced cancer patients and their families, to maximise quality of life and patient care.

## Figures and Tables

**Figure 1 jpm-12-00954-f001:**
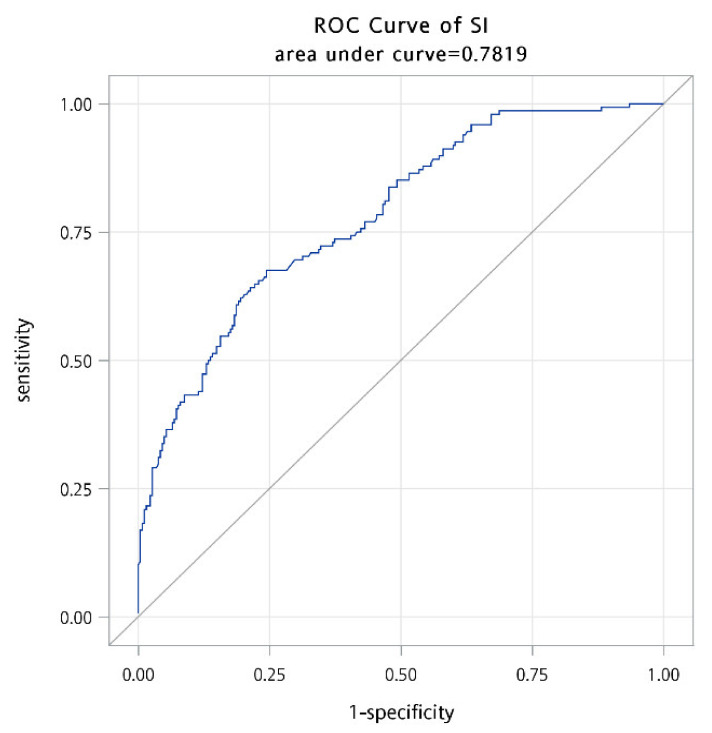
Receiver operating characteristic curve of Shock Index in predicting 60-day mortality.

**Figure 2 jpm-12-00954-f002:**
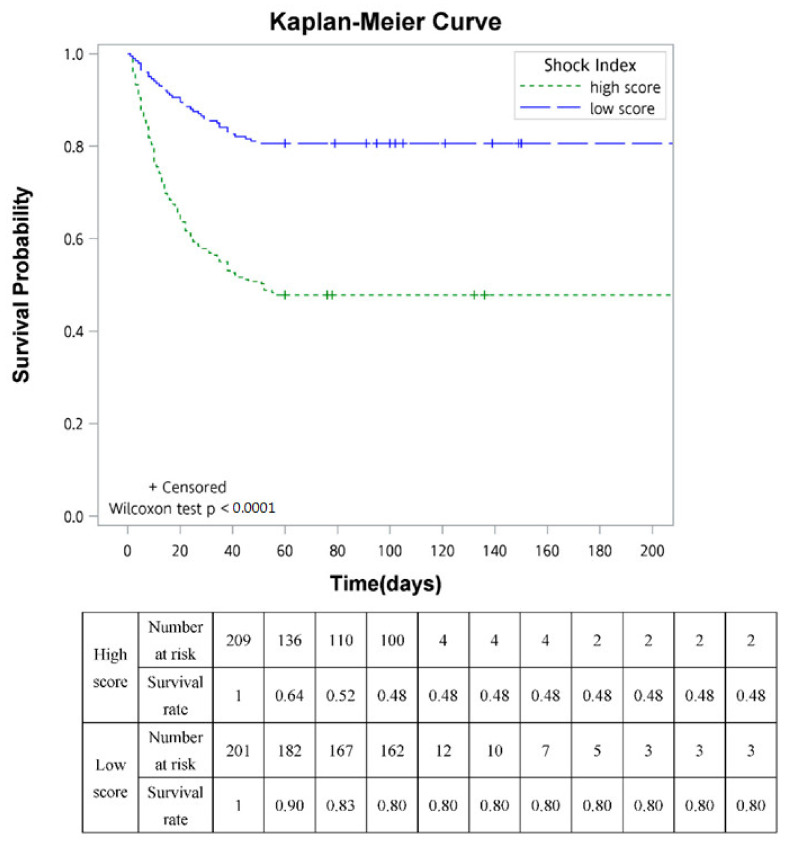
Kaplan-Meier curves of 60-day mortality for advanced cancer patients with SI> 0.94 (high score) and SI< 0.94 (low score).

**Table 1 jpm-12-00954-t001:** Comparison of the medical history of patients, survivors versus non-survivors at 60 days after the index emergency department visit.

Variable	Patients			*p*-Value
	Total	Survivors	Non-Survivors	
No. of Patients	410	258	152	
Age	63.25 ± 11.98	63.19 ± 11.92	63.36 ± 12.14	0.8947
Male (%)	250 (60.98)	160 (61.07)	90(60.81)	1.0000
**Primary cancer (%)**				
Thyroid cancer	2 (0.49)	2 (0.76)	0 (0)	0.7432
Hypo-pharyngeal cancer	9 (2.20)	4 (1.53)	5 (3.38)	0.2302
Lung cancer	130 (31.71)	89 (33.97)	41 (27.70)	0.2305
Oropharyngeal cancer	21 (5.12)	14 (5.34)	7 (4.73)	0.9701
Nasopharyngeal cancer	5 (1.22)	3 (1.15)	2 (1.35)	1.0000
Oesophageal cancer	20 (4.88)	12 (4.58)	8 (5.41)	0.8935
Gastric cancer	15 (3.66)	8 (3.05)	7 (4.73)	0.5522
Colon cancer	33 (8.05)	23 (8.78)	10 (6.76)	0.5935
Rectal cancer	14 (3.41)	10 (3.82)	4 (2.70)	0.7539
Bladder cancer	10 (2.44)	9 (3.44)	1 (0.68)	0.1596
Renal cancer	7 (1.71)	5 (1.91)	2 (1.35)	0.9830
Prostate cancer	7 (1.71)	6 (2.29)	1 (0.68)	0.4150
Cervical cancer	4 (0.98)	3 (1.15)	1 (0.68)	1.0000
Uterine cancer	2 (0.49)	2 (0.76)	0 (0)	0.7432
Ovarian cancer	1 (0.24)	1 (0.38)	0 (0)	1.0000
Brain cancer	6 (1.46)	6 (2.29)	0 (0)	0.1537
Pancreatic cancer	27 (6.59)	15 (5.73)	12 (8.11)	0.4672
Hepatic cell cancer *	35 (8.54)	14 (5.34)	21 (14.19)	0.0038
Gallbladder cancer	1 (0.24)	1 (0.38)	0 (0)	1.0000
Lymphoma	10 (2.44)	7 (2.67)	3 (2.03)	0.9417
Breast cancer	33 (8.05)	17 (6.49)	16 (10.81)	0.1751
Cholangial cancer	7 (1.71)	2 (0.76)	5 (3.38)	0.1173
Spinal cancer	1 (0.24)	0 (0)	1 (0.68)	0.7720
Tonsil cancer	2 (0.49)	2 (0.76)	0 (0)	0.7432
Melanoma	4 (0.98)	3 (1.15)	1 (0.68)	1.0000
Soft tissue cancer	4 (0.98)	4 (1.53)	0 (0)	0.3234
**Previous treatment (%)**				
Chemotherapy	286 (69.76)	177 (67.56)	109 (73.65)	0.2389
Radiotherapy	179 (43.66)	111 (42.37)	68 (45.95)	0.5497
Target therapy	74 (18.05)	44 (16.79)	30(20.27)	0.4560
Surgical treatment *	316 (77.07)	216 (82.44)	100 (67.57)	0.0009
**Comorbidities (%)**				
Diabetes mellitus	107 (26.10)	68 (25.95)	39 (26.35)	1.0000
Hypertension	162 (39.51)	106 (40.46)	56 (37.84)	0.6774
Cerebrovascular accident	25 (6.10)	17 (6.49)	8 (5.41)	0.8217
Heart failure	10 (2.44)	8 (3.05)	2 (1.35)	0.4594
Coronary artery disease	18 (4.39)	11 (4.20)	7 (4.73)	0.9990
Chronic obstructive pulmonary disease	19 (4.63)	12 (4.58)	7 (4.73)	1.0000
End stage renal disease	6 (1.46)	6 (2.29)	0 (0)	0.1537
Liver cirrhosis	34 (8.29)	16 (6.11)	18 (12.16)	0.0513
Bed-ridden status	9 (2.20)	5 (1.94)	4 (2.70)	0.8601

* denotes statistical significance.

**Table 2 jpm-12-00954-t002:** Comparison of the clinical findings of patients, survivors versus non-survivors at 60 days after the index emergency department visit.

Variable	Patient							
	Total	Survivors	Non- Survivors	*p*-Value	Univariate OR (95%CI)		Multiple OR ** (95%CI)	
No.	410	258	152					
Body temperature (°C) *	36.96 ± 1.09	37.08 ± 1.12	36.75 ± 0.99	0.0019	0.74	(0.61, 0.90)		
Pulse rate (/min) *	109.30 ± 22.54	106.80 ± 22.62	113.60 ± 21.82	0.0031	1.01	(1.00, 1.02)		
Respiratory rate (/min) *	21.06 ± 4.33	20.42 ± 4.01	22.20 ± 4.66	<0.0001	1.1	(1.05, 1.16)		
Systolic blood pressure (mmHg) *	117.80 ± 28.45	127.80 ± 27.21	100.10 ± 21.04	<0.0001	0.95	(0.94, 0.96)		
Diastolic blood pressure (mmHg) * Mean arterial pressure (mmHg) *	71.96 ± 16.75 87.32 ± 19.51	76.25 ± 16.76 93.43 ±19.06	64.36 ± 13.82 76.50 ± 15.15	<0.0001 <0.0001	0.95 0.94	(0.93, 0.96) (0.93, 0.96)		
Shock index *	0.98 ± 0.33	0.87 ± 0.24	1.19 ± 0.36	<0.0001	76.43	(28.00, 208.63)	1.591	(1.42, 1.78)

* indicates statistical significance. ** performed by logistic regression model adjusted for age, sex, personal medical and medication history.

**Table 3 jpm-12-00954-t003:** Optimal cut-off value for SI with corresponding accuracy, sensitivity, and specificity.

Cut-Off Point	Accuracy Rate	Sensitivity	Specificity	PPV	NPV
0.94	66.10%	73.65%	61.83%	52.15%	80.60%

## Data Availability

The datasets generated and analyzed during the current study are available from the corresponding author on reasonable request.

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
