# Peer review of "Shock Index Is a Validated Prediction Tool for the Short-Term Survival of Advanced Cancer Patients Presenting to the Emergency Department"

_jpm, 2022, doi:10.3390/jpm12060954_

Round 1
Reviewer 1 Report
The article is well written and suitable for publication in a journal
Author Response
Reviewer #1’s Comments
The article is well written and suitable for publication in a journal
Response to Reviewer #1
We thank the reviewer for the kind remarks.

Reviewer 2 Report
By its title and stated purpose, the manuscript presents this study as a validation of the shock index (SI) as a tool for predicting 60-day mortality. However, the present study is basically a replication of this group’s prior recent study of the SI (Ref. 18) and thus it would be better characterized as a replication, as opposed to a validation, study. The assertion that the present study is prospective whereas the prior study was retrospective is largely a distinction without a difference. The design, measurements, analytic strategy, and manner of presentation of results in tables and figures is remarkably similar in both studies. Indeed, the present study’s Abstract concludes that the present study validates the use of the SI as a predictive tool solely because its results are similar to (i.e., replicate) those of the prior study.
If the study is to be represented as a validation study, then the instrument (e.g., dichotomy of SI based on a specified cutoff), the type of validity, and the criterion by which the validity of the instrument will be evaluated ought to be specified a priori. If the study is to be represented as a replication study, then there ought to be detailed comparisons of methods and results between the original and the replicate study.
Author Response
Reviewer #2’s Comments
By its title and stated purpose, the manuscript presents this study as a validation of the shock index (SI) as a tool for predicting 60-day mortality. However, the present study is basically a replication of this group’s prior recent study of the SI (Ref. 18) and thus it would be better characterized as a replication, as opposed to a validation, study. The assertion that the present study is prospective
whereas the prior study was retrospective is largely a distinction without a difference. The design, measurements, analytic strategy, and manner of presentation of results in tables and figures is remarkably similar in both studies. Indeed, the present study’s Abstract concludes that the present study validates the use of the SI as a predictive tool solely because its results are similar to (i.e.,
replicate) those of the prior study.
If the study is to be represented as a validation study, then the instrument (e.g., dichotomy of SI based on a specified cutoff), the type of validity, and the criterion by which the validity of the instrument will be evaluated ought to be specified a priori. If the study is to be represented as a replication study, then there ought to be detailed comparisons of methods and results between the
original and the replicate study.
Response to Reviewer #2
Thank you for your pertinent suggestions to improve our manuscript. This study was designed to be a validation study – we first analysed the data as if it were a primary study (which revealed an optimal cut-off point of 0.99), followed by secondary validation of the cut-off point of 0.94 found in our previous study. Because the results were similar, we had initially opted to omit the evaluation of 0.94 cut-off to streamline the discussion section and avoid confusion among readers.
However, we acknowledge the reviewer’s valid concerns regarding the design of the study. We have therefore added the following sections to our manuscript.
Materials and Methods: 2.4. Statistical Analysis section, last paragraph [Page 3]
“Validation of our previous study’s cut-off point of 0.94 was further performed by evaluating its sensitivity, specificity, negative predictive value, positive predictive value, and accuracy in predicting 60-day mortality of advanced cancer patients presenting to the ED in this current study.”
Results section, last paragraph [Page 6-7]
“Validation of our previous study’s optimal cut-off point of 0.94 in this current study population found that it had a comparably good performance, with sensitivity 73.65%, specificity 61.83%, positive predictive value of 52.15%, negative predictive value of 80.60%, and accuracy 66.1%. Patients with
SIs > 0.94 had a hazard ratio of 3.442 compared to those with SIs < 0.94 (p<0.0001).”
Discussion section, 2nd paragraph, line 3-8 [Page 7]
“Further validation of our previous study’s cut-off point of 0.94 found that it was still able to identify 73.65% of patients who might benefit from early initiation of palliative care. Nevertheless, it must be heavily emphasized that SI should not be taken as the sole deciding factor of determining goals of therapy, but rather an adjunct to the ongoing conversation with the cancer patient and family about
their wishes regarding end-of-life care.”
Discussion section, last 2nd paragraph [Page 8]
“While the previous cut-off point of 0.94 has been validated in this current study, analysis of this current study population found a similar optimal cut-off point of 0.99. Given that the latter was based on a larger study population over a longer study period, in addition to considerations of ease of calculation, we are of the opinion that an SI of 0.99 would prove more applicable in the ED clinical
setting.

Reviewer 3 Report
Shock index is a valuable prognostic tool with respect to patients with advanced cancer presenting in an emergency department. Shock index as the ratio of pulse rate to systolic blood pressure is a very simple measure. It is of value in predicting the probability of a patient to survive for 60 days or longer. In a stepwise multivariable analysis, it remained as the only independent prognostic variable.
There are a few points which should be modified in the manuscript.
- It should be stressed that life-sustaining interventions produce a burden of huge hospital bills ONLY in countries with insufficient health insurance systems.
- In the Kaplan Meier curves, a risk table should be added.
- There is no description of the type of treatment that study patients have received. Has actual treatment been already been informed by SI? Has actual survival been influenced by the choice of treatment (life-sustaining measures versus palliative care)? This might be a strong limitation of the study and should be discussed accordingly.
Author Response
Reviewer #3’s Comments
Shock index is a valuable prognostic tool with respect to patients with advanced cancer presenting in an emergency department. Shock index as the ratio of pulse rate to systolic blood pressure is a very simple measure. It is of value in predicting the probability of a patient to survive for 60 days or longer.
In a stepwise multivariable analysis, it remained as the only independent prognostic variable.
There are a few points which should be modified in the manuscript.
ï‚· It should be stressed that life-sustaining interventions produce a burden of huge hospital bills ONLY in countries with insufficient health insurance systems.
ï‚· In the Kaplan Meier curves, a risk table should be added.
ï‚· There is no description of the type of treatment that study patients have received. Has actual treatment been already been informed by SI? Has actual survival been influenced by the choice of treatment (life-sustaining measures versus palliative care)? This might be a strong limitation of the study and should be discussed accordingly.
Response to Reviewer #3
Question 1: It should be stressed that life-sustaining interventions produce a burden of huge hospital bills ONLY in countries with insufficient health insurance systems.
Answer 1: We have added the following to our manuscript [Introduction section, 1st paragraph, lines 8-9 (Page 2, lines 8-9)]
“Advanced cancer patients consequently face increased suffering for the remainder of their lives while their families are saddled with the financial burden of huge hospital bills [4,5]; this occurrence is especially true in countries with limited health insurance systems.”
Question 2: In the Kaplan Meier curves, a risk table should be added.
Answer 2: We have added a risk table to the Kaplan-Meier curve as suggested [Results section: Figure 2 (Page 6)]
Question 3: There is no description of the type of treatment that study patients have received. Has actual treatment been already been informed by SI? Has actual survival been influenced by the choice of treatment (life-sustaining measures versus palliative care)? This might be a strong limitation
of the study and should be discussed accordingly.
Answer 3: This was an observational study to validate our previous results. As such, the evaluation of patients with SI did not interfere with the treatment of patients. All recruited patients received prompt treatment for their respective presenting illnesses. We have clarified accordingly in our manuscript
[Materials and Methods: 2.2. Setting and Subjects section, lines 2-5 (Page 2)]
“All adult advanced cancer patients above the age of 18 years who visited the ED of our hospital from January 2019 to April 2021 were invited to participate in this study, with the explicit understanding that the research data obtained would not be used to influence decisions on management options and goals. All patients received prompt treatment for their respective presenting illnesses as per our ED protocols.

Round 2
Reviewer 2 Report
I appreciate that the manuscript now includes information that evaluates/validates the previous study’s optimal cut-off point of 0.94 in the current study population. Nevertheless, the manuscript is written primarily as if this new study was a primary study, not a validation study. It presents a new optimal cut-off point of 0.99 as superior to the prior one, as if the results from the current study are more valid than those of the prior study.
Author Response
Reviewer #2’s Comments:
I appreciate that the manuscript now includes information that evaluates/validates the previous study’s optimal cut-off point of 0.94 in the current study population. Nevertheless, the manuscript is written primarily as if this new study was a primary study, not a validation study. It presents a new optimal cut-off point of 0.99 as superior to the prior one, as if the results from the current study are more valid than those of the prior study.
Response to Reviewer #2:
We thank the reviewer for pointing out further areas of improvement for our
manuscript. We have removed all mention of 0.99 as a “new optimal cut-off point” and focused solely on validating 0.94 as our cut-off point. We hope this streamlines our discussion and validation of 0.94 shock index in predicting 60-day mortality of advanced cancer patients presenting to the ED.
Multiple amendments have been made to the manuscript, highlighted in GREEN.
Materials and Methods: 2.4. Statistical Analysis section, line 5-8 [Page 3]
Multivariable logistic regression was subsequently done to obtain the odds ratio with respect to 60-day mortality, and receiver operator calibrating (ROC) curve of this study population was plotted. Validation of our previous study’s cut-off point of 0.94 was performed via evaluation of its sensitivity, specificity, negative predictive value, positive predictive value, and accuracy in this current study population. Kaplan-Meier analysis was also employed to examine survival between groups with high versus low SIs. P-values of <0.05 were taken to be statistically significant.
Results section, last 2nd paragraph [Page 5]
Validation of our previous study’s optimal cut-off point of 0.94 in this current study population found that it had a comparably good performance, with sensitivity 73.65%, specificity 61.83%, positive predictive value of 52.15%, negative predictive value of 80.60%, and accuracy 66.10% (Table 3). Patients with SIs > 0.94 had a hazard ratio of 3.442 compared to those with SIs < 0.94 (p<0.0001).
Results section, Table 3 [Page 5]:
Table 3. Optimal cut-off value for SI with corresponding accuracy, sensitivity, and specificity.
Cut-off point Accuracy rate Sensitivity Specificity PPV NPV
0.94 66.10% 73.65% 61.83% 52.15% 80.60%
Results section, Figure 2 [Page 6]: (please refer to the attached file)
Discussion section, 3rd paragraph [Page 7]
The beauty of SI lies in its simplicity of calculation, based on two vital sign
measurements which can be rapidly obtained in less than a minute. With an optimal cut-off point of 0.94, clinicians in the ED should consider discussing with advanced cancer patients and their families regarding the option of palliative care once they see that pulse rate readings are almost equal to or higher than the corresponding systolic blood pressures
